# Bioactivity Comparison of Electrospun PCL Mats and Liver Extracellular Matrix as Scaffolds for HepG2 Cells

**DOI:** 10.3390/polym13020279

**Published:** 2021-01-16

**Authors:** Igor Slivac, Emilija Zdraveva, Fran Ivančić, Bojan Žunar, Tamara Holjevac Grgurić, Višnja Gaurina Srček, Ivan-Krešimir Svetec, Tamara Dolenec, Emi Govorčin Bajsić, Mirna Tominac Trcin, Budimir Mijović

**Affiliations:** 1Faculty of Food Technology and Biotechnology, University of Zagreb, Pierottijeva 6, 10000 Zagreb, Croatia; islivac@pbf.unizg.hr (I.S.); ivancic.fran@gmail.com (F.I.); bzunar@pbf.unizg.hr (B.Ž.); vgaurinasrcek@pbf.unizg.hr (V.G.S.); iksvetec@pbf.unizg.hr (I.-K.S.); 2Faculty of Textile Technology, University of Zagreb, Prilaz baruna Filipovića 28a, 1000 Zagreb, Croatia; emilija.zdraveva@ttf.hr; 3Faculty of Metallurgy, University of Zagreb, Aleja narodnih heroja 3, 44000 Sisak, Croatia; tholjev@simet.hr; 4Department of Transfusion and Regenerative Medicine, University Hospital Centre Sestre Milosrdnice, Draškovićeva 19, 10000 Zagreb, Croatia; tamara.dolenec@gmail.com; 5Faculty of Chemical Engineering and Technology, University of Zagreb, 10000 Zagreb, Croatia; egovor@fkit.unizg.hr; 6Institute of Immunology, Rockefellerova 2, 10000 Zagreb, Croatia; mirna.tomtrcin@gmail.com

**Keywords:** PCL, extracellular matrix, ECM, HepG2 cells, electrospinning

## Abstract

Cells grown on bioactive matrices have immensely advanced many aspects of biomedical research related to drug delivery and tissue engineering. Our main objective was to perform simple evaluation of the structural and biotic qualities of cell scaffolds made of affordable biomaterials for liver cell line (HepG2) cultivation in vitro. In this work the electrospun matrix made of synthetic polyester poly(ε-caprolactone) (PCL) was compared with the natural protein-based extracellular matrix isolated from porcine liver (ECM). Mechanical and structural analysis showed that ECM was about 12 times less resistant to tensile stress while it had significantly larger pore size and twice smaller water contact angle than PCL. Bioactivity assessment included comparison of cell growth and transfection efficiency on cell-seeded scaffolds. Despite the differences in composition and structure between the two respective matrices, the rate of cell spreading and the percentage of transfected cells on both scaffolds were fairly comparable. These results suggest that in an attempt to produce simple, cell carrying structures that adequately simulate the natural scaffold, one can rely on PCL electrospun mats.

## 1. Introduction

The key issue of tissue engineering and certain aspects of life sciences is combining living cells with biocompatible scaffolds and constructing a template for stable cell growth. There are several critical points in this process and often the initial one is the choice of biomaterial(s) to be used as cell scaffolds. The role of biomaterial is to provide physical support for engineered tissue-like or 3D cell construct and allow adequate micro-environmental cues for cell proliferation. Besides sustaining cell adhesion and multiplication, bioactive scaffolds should ensure structural porosity, controlled degradation and mechanical resistance against the pressure exerted at both in vitro and in vivo conditions. In the past two decades, numerous biomaterials, mostly polymers, have been used to prepare scaffolds for tissue engineering including natural materials, such as collagen, alginate and chitosan; or synthetic materials, such as poly(ε-caprolactone) (PCL), polyglycolic acid and polylactic acid. Furthermore, in recent years a vast number of composite polymer-based matrices as well as bio-functionalized polymers have been investigated for improved biomedical properties [1,2]. Alternatively, extracellular matrix (ECM) has been extensively studied as the most natural cell scaffold for tissue and organ regeneration. ECM is a heterogeneous network of fibrous glycoproteins serving as physical platform for coordination of cell proliferation and differentiation, and it is produced by the tissue’s own cells. Isolated by decellularization process, and seeded with suitable cells, ECM has proved itself as an inductive template for functional tissue renewal in skin, bone, nerve, heart, lung, liver, kidney, bladder and other organs [3]. Development of confluent cell constructs on porous and nano-fibrous scaffolds serves as a promising strategy for in vitro drug and cancer studies.

In this work we compared properties of synthetic electrospun and natural scaffolds for sustaining growth of a liver cell line (Hep G2). The scaffolds tested were PCL electrospun matrix and naturally complex extracellular matrix isolated from pork liver (ECM). PCL is aliphatic polyester composed of repeated units of hexanoate. Although its biocompatibility is rather low, its physical properties and adjustable biodegradability, including ease of blending and copolymerisation, make it a desirable cell scaffold material as well as drug delivery particles [4,5,6]. The major limitation of PCL matrices is hydrophobicity, which might affect cell adhesion efficiency and duration of biodegradability. However, this can be easily overcome by alkali treatment or blending with more hydrophilic components. In this paper PCL mats were produced by electrospinning, a method that uses electric force to extract polymeric threads out of monomer solution and makes fiber network on collector surface. Electrospinning is fast developing fiber production platform ranging from single-fluid blending process [7], solid needle process [8], coaxial [9], side-by-side [10], tri-axial [11] and some other complex processes [12]. However, the single-fluid strategies are most widely applied in large-scale production due the simplicity of implementation. Intensity of electric field, flow rate of polymer solution, as well as needle diameter and collector design are critical factors for fiber morphology and functional properties of produced scaffolds [13]. One of the most attractive features of elctrospinning in our research is the ability to produce fiber-based scaffolds similar to the fibrous structure of ECM. Depending on set-up parameters, it enables production of matrices of specific physical properties and it is comparatively very cost efficient. Hepatic porcine ECM was obtained by removing endogenous cellular components from a native porcine liver through the process of decellularization. Various protocols have been available for decades and can include mechanical, freeze/thaw, enzymatic, detergent, and solvent treatments. The goal is always the same: getting biochemically and structurally preserved ECM. We have chosen the most common one, using a nonionic surfactant Triton X-100 [14,15]. Porcine organ and tissue decellularization has high relevance in translational medicine, compared to other species, due to its availability and high similarities of its biochemical profile with that of humans [16,17]. As mentioned earlier, the cell line used for our scaffold evaluation was HepG2. The cells originate from human liver cancer tissue and exhibit epithelial-like morphology growing as monolayer in small aggregates. However, they are not tumorigenic and have high proliferation rates. Moreover, they are reliable and economical substitute for primary or stem cells, and hence are commonly used in early hepatoxicity and in vitro liver regeneration studies [18,19].

Rather than promoting novel and sophisticated materials, our research contributes to revalidation of simple and affordable biomaterial (PCL) through comparative bioactivity assessments with natural cell scaffold (ECM).

## 2. Materials and Methods

### 2.1. Production and Preparation of Scaffolds

In this work we used two materials to produce fibrous scaffolds for cell immobilization: poly(ε-caprolactone) (PCL), and natural extracellular matrix (ECM). PCL scaffolds were made by electrospinning while ECM-based scaffold was prepared from pork liver. For preparation of electrospun scaffolds we used the following materials: PCL, Mn = 80,000 (*Lach-ner*) and solvents: glacial acetic acid and acetone (Kemika, Croatia). PCL scaffolds were prepared from PCL polymer solution 18 % (*w/v*) by dissolving the polymer in glacial acetic acid and acetone (*v/v* 8:2) with constant stirring for more than 24 h. PCL solution was electrospun on the electrospinning device NT-ESS-300, NTSEE Co. Ltd. South Korea. The process parameters were set as following: electrical voltage of 15–17 kV, needle tip to collector distance of 18 cm, volume flow rate of 1 mL/h and electrospinning time of 4 h. The electrospinning was conducted through a BD plastic syringe, with a blunt needle of 21G, and inner and outer diameter of 0.82 and 0.51 mm respectively. A flat metal plate served as the PCL fibers collector. Before cell seeding, the electrospun PCL sheets were cut into discs fitting the size of wells in 24-well plate. The discs were distributed in the plate where they were cleaned in 70% ethanol and exposed to UV light for 30 min, soaked in 1M NaOH for one hour and finally washed three times in PBS (Sigma, UK) and conditioned in culture medium with serum.

Production of liver ECM was carried out following a slightly modified decellularization protocol with non-ionic surfactants reported in [14,20]. Briefly, a fresh piece of pork liver was kept frozen at −80 °C overnight. The next day approximately 2 mm thick tissue bands were cut with mechanical meat cutter. The bands were then trimmed into discs about 2 cm in diameter and placed in a 1 L bottle with decellularization solution (1% Triton X-100). The ratio of liver tissue to decellularization solution was 1:10 (*w/v*). The bottle was shaken on an orbital shaker (MRC Scientific instruments, Holon, Israel) at 200 rpm and 4 °C. The solution was changed approximately 4 times per day during two days. On the third day decellularized ECM was washed in deionized water (with antibiotic added) for the following 24 h, with periodic water change. Prior to cell seeding the ECM discs were placed in 24-well plate, soaked in 70% ethanol under UV light for 30 min, washed in PBS and conditioned in culture medium containing serum.

### 2.2. Physical Characterization of Scaffolds

The electrospun and the porcine ECM scaffolds had their structural and physical qualities evaluated. This involved estimation of fiber diameter (only for electrospun PCL scaffolds), pore area and total porosity, as well as tensile properties. The surface morphology of all matrices was observed under scanning electron microscopy SEM FEG QUANTA 250 FEI. The samples were not coated prior imaging and the images were taken at 1000× magnifications. The fiber diameter and pore area within scaffolds were estimated after 100 random measurements using open-source image processing program *ImageJ*, for SEM images (PCL scaffolds) or digital camera photos (ECM scaffolds). The scaffold thickness was measured with a digital micrometer, Digi Micrometer Mitutoyo (Mitutoyo, IL, USA). For that purpose ethanol-soaked and subsequently air-dried ECM samples were used, which ensured adequate stiffness of the material to be measured. Total porosity *p* of our scaffolds was calculated according to Equation (1) where *m* is sample weight, *A* is pore area, *h* is sample thickness and *ρ* is polymer density. The tensile tests were performed on *Statimat M*, *Textechno* tensile testing instrument with a load cell of 100 N, rate of 10–25 mm/min and gauge length of 75 mm. For pore size and the tensile strength determination ECM scaffolds were in wet state. Scaffolds surface wettability was evaluated through water contact angle measurement by the *ImageJ* Drop Analysis LB-ADSA tool from the images captured by Dino Capture 2.0 microscope (Dino-Lite, Almere, The Netherlands). All tested scaffold samples were in dry state.
(1)$$P=\left(1-\frac{m}{A\cdot h\cdot \rho }\right)\cdot 100$$

### 2.3. Cell Seeding and Cell Cultivation on Scaffolds

In this study HepG2 cell line was used (ATCC HB-8065). It was grown in Dulbecco’s Modified Eagle Medium (Capricorn Scientific GmbH, Germany) with 10% fetal bovine serum (Sigma), and 1% antibiotic (Antibiotic-Antimycotic 100×, Gibco, Thermo Fisher Scientific, Loughborough, UK). While growing in Petri-dish or seeded on scaffolds in 24-well plates, the cells were incubated at 37 °C in a humidified CO_2_ incubator. The cell inoculum for seeding the scaffolds was harvested form a Petri-dish using 0.25% Trypsin-EDTA (Sigma, St. Louis, MO, USA) and set at 200,000 cells per well (i.e., per scaffold). An estimation of inoculum density and viability was carried out using trypan-blue method in Neubauer hemocytometer. The plates with seeded scaffolds were gently put inside the incubator for cells to attach. After six hours, samples of each scaffold type were taken for comparison of cell attachment efficacy. The remaining scaffolds were transferred in a new 24-well plate with fresh culture media (0.5 mL/well) for regular monitoring of cell growth dynamics. During the first four days, media was changed daily and later on twice a day. The cell seeding and spreading on scaffolds were tracked using standard 3-(4,5-dimethylthiazol-2-yl)-2,5-diphenyltetrazolium bromide (MTT) (Invitrogen, OR, USA) assay 48, 96 and 144 h post seeding. The assay is based on conversion of water-soluble tetrazolium salt into insoluble dark blue formazan crystals by metabolically active cells. The crystals can be dissolved in organic solvent (dimethyl-sulfoxide, DMSO, Kemika, Croatia), and the color intensity of the solution, determined simply by spectrophotometry at 570 nm, is relative to the actual cell quantity. Briefly, the scaffolds (in sextuplicate) were transferred into a new 24-well plate, incubated at 37 °C in 1 mL media with 5% MTT reagent. Four hours later each scaffold was transferred in 2 mL DMSO and shaken for 15 min at room temperature to dissolve the formed formazan. Absorbance values were measured using spectrophotometer (Thermo Scientific Genesys 10S, Bremen, Germany). Unseeded scaffolds were treated the same as the seeded ones and the obtained DMSO used as blank.

### 2.4. Cell Transfection

The cell transfection was carried out with plasmid DNA (pDNA) containing *mCherry* as the reporter gene under regulation of cytomegalovirus CMV promoter. The plasmid was constructed by ligation of 5.3 kb EcoRI-SacI fragment of the plasmid pcDNA3.1/His/LacZ (Invitrogen Life Technologies, Carlsbad, CA, USA) and 0.7 kb EcoRI-SacI fragment of the plasmid mCherry-pBAD. DNA manipulations and restriction cloning were performed as in [21], using *Escherichia coli* DH5α [22]. *TurboFect^TM^* (Thermo Scientific) was used as HepG2 cell transfection reagent. Scaffolds plated into 24-well plates were seeded with 100,000 cells per well. The control wells were seeded with 33,000 cells. These cells grew on the plastic surface of the cultivation plate. All samples were done in triplicate. Two days later the cell-seeded scaffolds were transferred into new wells with fresh media. For cell transfection, pDNA–Turbofect complexes were added into the culture media at final pDNA concentrations 1 µg/mL. For complexation with pDNA, *Turbofect^TM^* was diluted according to manufacturer’s protocol. Transfection optimization was made previously on monolayer HepG2 culture. The applied ratio of DNA solution and transfection agent was 1:4 (v:v). The cells were cultured in the presence of complexes for 5 h in CO_2_-incubator and then the final media replacement was done in all the wells. The transfection efficiency was analyzed after 48 h by registering red fluorescent protein-positive cells by means of inverted fluorescence microscopy at 400× magnification (EVOS Floid Cell Imaging Station, Life Sciences). Prior to microscopy standard fluorescein-diacetate staining was carried out in order to make viable cell population visible on the non-transparent scaffolds. Fluorescence images of cells with fluorescein and red mCherry protein were registered in appropriate channels. Fluorescence thresholds were adjusted manually to distinguish between transfected and untransfected cells. Quantification of both red and green fluorescent cells was carried out in 24-well plate with the aid of an open source image-processing program ImageJ. Briefly, 10 fields of view with approximately 100 cells in each field were scanned in each well. The transfection efficiency (expressed as %) was calculated from the number of red fluorescent protein-expressing cells versus viable cell number (i.e., green + red cells).

### 2.5. Statistical Analysis

All experiments regarding physical characterization and bioactivity of matrices were performed in sextuplicate, unless stated differently. The data are expressed as the mean ± standard deviation. Analysis of variance (ANOVA one-way) and Tukey post-tests were performed in Origin8 to compare means between groups and were considered significantly different at the level of 0.05.

## 3. Results

### 3.1. Physical Characterization of the Scaffolds

The design of organized structures that mimic ECM of native tissues is important for rapid tissue regeneration as well as for obtaining suiTable 3D tissue models for in vitro drug and cancer research [23,24]. One of the most important assets expected from a potential tissue scaffold is bioactivity. It ensures successful cell growth and the required biological response of the desired tissue analog. Electrospinning techniques offer the possibility to tailor scaffolds of various chemical and physical properties as well as topography [25]. The PCL mats used in this work were electrospun with the same parameters applied in our previous work with primary cells [6]. To obtain ECM scaffolds, decellularization of pork liver tissue was carried out using slightly adapted protocol described in [8,14]. 

Both scaffolds were of similar thickness in dry state, ranging between 0.5 and 1 mm. They however exhibited significant structural difference, both macroscopically and microscopically, as shown in Figure 1. The PCL mats were smoother, more compact and resilient than ECM scaffolds. They also had much more uniform texture resulting from condensed, mostly uniform, cylindrical fibers that gave the scaffold its consistency but less permeability and porosity compared to the liver ECM. The typical composition of ECM includes almost entirely high-molecular weight proteins and some glycan, which, due to their hydrophilic properties, make the scaffold swell in aqueous environment and gain elasticity [26]. 

Using the images from SEM (for PCL) and standard light microscopy (for ECM) fiber diameter and pore area distribution were estimated. For the PCL scaffolds, the range of the fiber diameter was from ~100 nm to 2.8 µm, with a mean diameter of 874 nm (Table 1). Random distribution of thin PCL fibers affected the pore size as well as the total scaffold porosity (Figure 2A, Table 1). ECM scaffolds had evidently very large pores compared to PCL mats, often more than two orders of magnitude larger. Their average pore size was 785.5 µm^2^. Due to the ECM structure, and our technical limitations, we were not able to evaluate the fiber diameter or total porosity of the scaffolds. However, according to some recent studies, the scaffold porosity seems to have little effect on cell differentiation or migration. Besides stiffness, it is the pore size that plays the major role in in vitro cell maturation and post-seeding distribution [27]. 

Net cell displacement is best achieved within matrices with cell-size pores. In our case the electrospun PCL matrix had micro-sized pores and thus often exhibited slightly, but not significantly, higher cell confluence than the liver ECM with hundredfold larger pores (Figure 2A and Figure 3). As for the integrity and strength of our scaffolds, the results are shown in Figure 2B. It is important to note that the tensile strength of the liver ECM was more than 10 times lower than that of the PCL mats, which was expected due to the large size of ECM pores (Table 1). 

In general, soft tissues have peculiar mechanical characteristics, which still present technical challenges to be solved by the experts. For instance, tissue specimens are usually small dissecting samples omitting the effect of the tissue integrity and connections. Moreover, mechanical tests are usually performed under different humidity and temperature conditions, with different forces applied [28]. Considering the required resistance to mechanical stress forces occurring in a soft tissue that does not directly partake in body motion (like liver, skin, ocular tissue etc.), the tensile strengths of both of our scaffolds are within satisfactory limits, despite their twelvefold difference [14,29].

The measurement of wettability represents an important contribution in the evaluation of biomaterials. When the water contact angle on a biomaterial surface is greater than 90°, the surface is usually referred to as hydrophobic. The wettability of our scaffolds evaluated after measuring water drop contact angles is given in Figure 2C. The hydrophobicity of the neat electrospun PCL mats was rather high, regardless of their pre-treatment in NaOH solution. In contrast to the PCL mats, the surface of liver ECM was pretty hydrophilic, as evident from the very low contact angle (~61°). It is well known that high surface wettability stimulates cell attachment and spreading, especially in the formation of early non-specific cell-surface bonds [30]. Therefore, our results suggest that ECM has more favorable cell adhesion surface, but this was not confirmed in our later experiments with HepG2 cells.

### 3.2. HepG2 Cell Seeding and Cultivation on the Scaffolds

After sterilization in ethanol and conditioning in growth medium, the scaffolds were ready for seeding. A dense HepG2 cell suspension was added dropwise to the scaffolds plated in 24-well plates. Six hours later, the first scaffold samples were taken to check the cell seeding efficiency. For tracking the cell growth dynamics and cell spreading over the scaffolds, the samples were taken every 48 h for 6 days. All samples were treated with tetrazolium salt which gets converted into bluish formazan by viable cell mitochondria (MTT assay). The spectrophotometric measurement of formazan absorbance showed the (relative) change in the quantity of living cells occupying the scaffolds (Figure 3). Although cell growth rate is hard to estimate without accurate cell counting, our early attempts to detach cells enzymatically and then quantify them under the microscope yielded very inconsistent results, especially for ECM. To finally get the comparable data, we pursued with the widely accepted colorimetric MTT method. Adherent cells such as HepG2 are very sensitive concerning growth surface properties especially at the initial attachment stage. However, in terms of seeding efficiency, our results showed that the cells had no significantly favorable scaffold. The number of cells attached was generally much lower than expected, considering the applied quantity of seeding cell. This could be particularly critical point if one aims to grow primary cells that are known for their limited growth capacity. As shown in Figure 3, both scaffolds initially have rather low cell density, with cells mostly distributed at the edges. Interestingly, in this and all later cell growth tests, the absorbance values for multiple ECM samples showed the highest variability, which might be explained by the pore size and macrostructural complexity of this scaffold. The overall low cell attachment could be explained by the imperfection of the chosen seeding method, considering the fact that the scaffolds were not well fixed at the bottom of the wells and consequently a certain amount of the cells could slip under scaffolds after seeding. In our future experiments we will try to prevent this by anchoring the disc shaped matrices with metal rings or certain non-toxic adhesives [31]. This shortcoming of the cell seeding technique was actually the main reason why we performed all the absorbance assays in sextuplicate. Judging from the visual inspection of the MTT-treated scaffolds, nearly complete cell confluence was achieved at day 6, especially on PCL scaffolds. It also appeared that the cells there had a more uniform growth rate, as suggested by the shorter deviation bars.

### 3.3. Transfection of HepG2 Cells Growing on the Scaffolds

The validation of any cell growth-supporting material cannot be considered complete without some sort of bioavailability assessment. Therefore, our bioactivity study included investigation of transfection efficiency of HepG2 cells. Immobilized multilayered cell culture, especially on natural ECMs or hydrogels with collagen, provides a more physiologically relevant system for studying normal and malignant human tissues than monolayer cultures [32]. This strategy proved advantageous in many cases in predicting the capacity of a tumor, or any other tissue, to respond to particular therapeutic regimes including transgene delivery [33]. In order to compare cell transgene uptake and marker protein production, we transfected HepG2 cells growing on our scaffolds and as monolayer in regular well-plates. The latter served as control. Cell seeding concentration was three times higher for scaffolds than for the control, considering the relatively low attachment efficacy previously described. However, the pDNA and transfection reagent quantity was kept the same for both conditions. To estimate viable cell density on the scaffolds, we treated the cells with fluorescein-diacetate. This allowed us to distinguish the population of transfected cells expressing mCherry from the untransfected ones using fluorescent microscopy. Image processing software was used for quantification of transfectants (Figure 4). Quantification of red and green cells enabled us to estimate the transfection efficiency. Overall, the number of transfected cells was rather low. The highest transfection efficiency was in the control group with about 42% of transfectants. In contrast, both tested scaffolds had almost half as many transfectants as the monolayer growing cells. Although there are many factors that may influence transfection efficiency, it is understood that nuclear translocation of pDNA occurs faster in actively dividing cells, such as HepG2 than in non-dividing cells because the nuclear membrane breaks down and reforms during cell division [34]. However, the sudden change of microenvironment given by fibrous matrices may slow down division rate of cells used to continuous plating in flat surface labware. Moreover, the transfection complex is more conveyable into cells grown as monolayer on flat surfaces than into cell aggregates or cells attached on 3D scaffolds [3]. In the case of ECM, it is likely that the charged polymer-pDNA compound got trapped in a hydrophilic ambience of proteins [35]. Applying a transfection agent of different chemistry, possibly lipid-based, might lead to improved transgene uptake. All these facts could partially explain the better transfection result observed in the control cell population, but at the same time it shows that the transport of biomolecules in cells growing on uneven and porous matrices is a much less successful process.

## 4. Conclusions

Synthetic matrices made of PCL have been the focus of biomedical research for about two decades. In this work we present the bioactivity comparison of electrospun PCL matrices with natural tissue scaffold isolated from porcine liver (ECM), by preparing 3D HepG2 cell constructs for in vitro studies. The different micro- and macrostructure, low hydrophilicity and significantly higher tensile strength of the PCL-based scaffolds did not play critical roles in HepG2 cell performance. The cells attached and grew well on both scaffolds, but showed no obvious matrix preference in terms of growth rate and transfection efficiency. On the other hand, the similar number of growing and RFP gene expressing cells indicate that PCL-based scaffolds successfully mimic the bioactivity found in ECM. Both PCL and ECM scaffolds have been thoroughly investigated over the past years, but it was rarely done by direct comparison of the two. Our work reconfirms that cell growth-supporting microenvironment could be engineered with affordable components and techniques, since electrospun PCL mats exhibit performance very similar to original tissue matrices. This finally means that in certain investigations PCL has potential to replace often inconsistent or unavailable ECM as a cell growth-supporting 3D substrate.

## Figures and Tables

**Figure 1 polymers-13-00279-f001:**
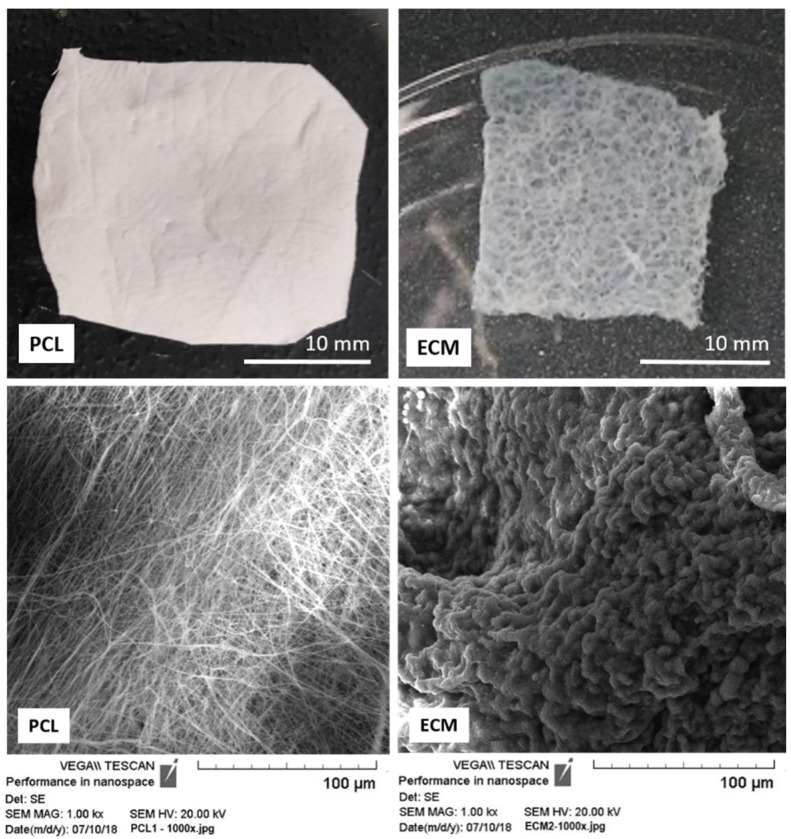
Scaffold samples prepared for comparative bioactivity assessment: electrospun PCL mat and porcine liver ECM. In the lower row are images of the scaffolds taken by SEM.

**Figure 2 polymers-13-00279-f002:**
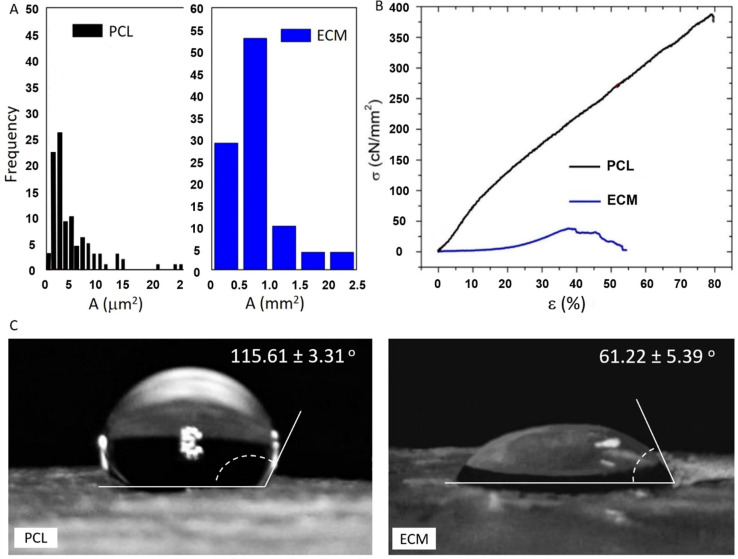
(**A**) Pore area distribution in the electrospun PCL mats and porcine ECM used as cell attachment scaffolds; (**B**) stress curves of tensile strength tested PCL mats and porcine ECM (*σ*-tensile strength, *ε*-strain); (**C**) checking the wettability by letting droplets of water for 30 s on the scaffold surface.

**Figure 3 polymers-13-00279-f003:**
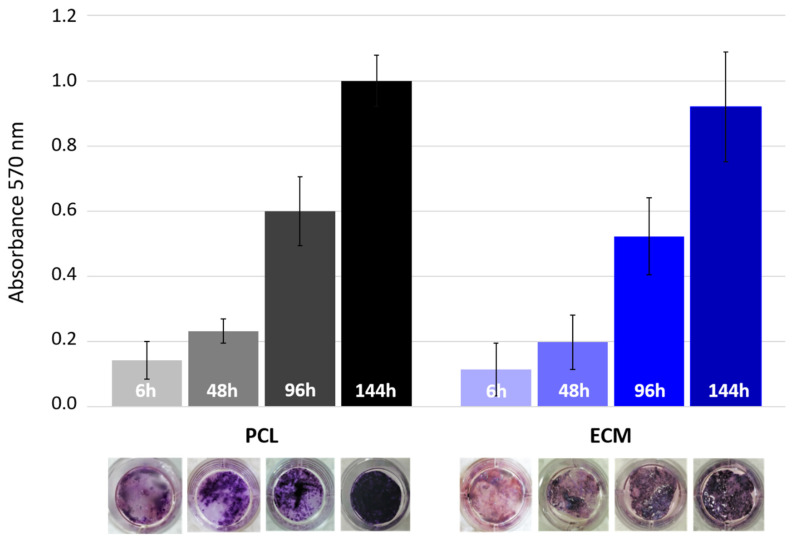
Growth of HepG2 cells on PCL and ECM scaffolds. MTT assay was used to track visually the cell confluence on scaffolds, and compare the cell growth rate during 144 h cultivation period by measuring dissolved formazan absorbance (bar chart).

**Figure 4 polymers-13-00279-f004:**
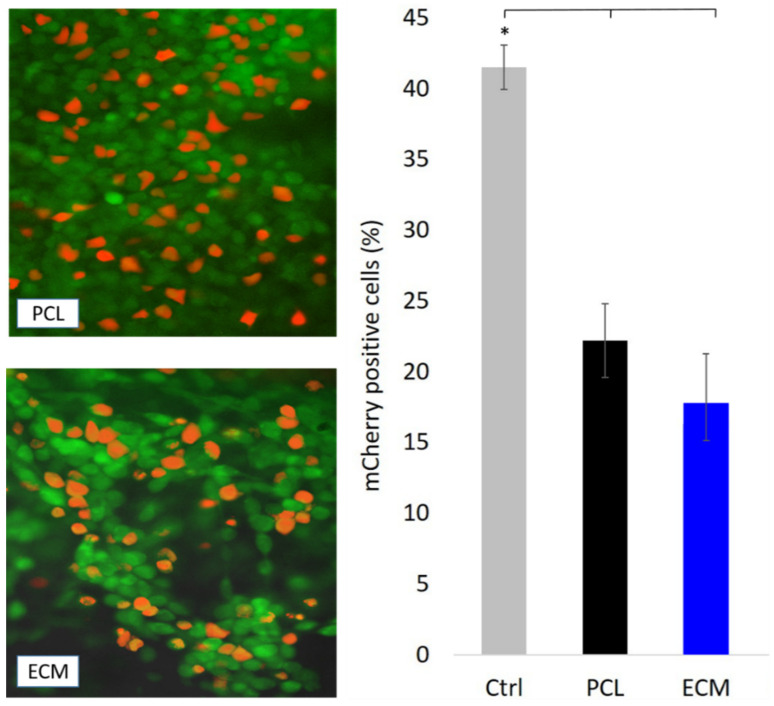
Transfection efficiency of HepG2 cells growing on PCL and ECM scaffolds. The cells were transfected with plasmid carrying red fluorescent protein (RFP) gene. Fluorescein-diacetate was applied to visualize the viable cells (green). Red and orange cells are producers of RFP (left), i.e., successfully transfected cells. Transfection efficiency of cells growing as monolayer on culture dish surface (Ctrl) and cells growing on the scaffolds is presented with bar chart (* *p* value < 0.05). There was no significant difference in the number of transfectants between two tested scaffolds.

**Table 1 polymers-13-00279-t001:** Structural and physical properties of assessed poly(ε-caprolactone) (PCL) and extracellular matrix (ECM) scaffolds: mean fiber diameter (*d_fibre_*), mean pore area (*A_pore_*), mean sample weight (*m*), mean sample thickness (*h*), polymer density (*ρ*), total porosity (*p*), maximum load (*F*), strain (*ε*) and tensile strength (*σ*).

Scaffold	*m* (g)	*h* (cm)	*ρ* (g/cm^3^)	*d_fibre_* (µm)	*A_pore_* (µm^2^)	*p* (%)	*F* (cN)	*ε* (%)	*σ* (cN/mm^2^)
PCL	0.079 ± 0.034	0.064 ± 0.017	0.663	0.874 ± 0.402	6.126 ± 2.899	81.4	592.67	86.63	383.18
ECM	0.051 ± 0.006	0.093 ± 0.036	0.295	-	785.5 ± 27.6	-	640	66.28	32

## Data Availability

The data presented in this study are available on request from the corresponding author.

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
