# Peer review of "Bioactivity Comparison of Electrospun PCL Mats and Liver Extracellular Matrix as Scaffolds for HepG2 Cells"

_polymers, 2021, doi:10.3390/polym13020279_

Round 1

Reviewer 1 Report

The manuscript compares the properties of electrospun PCL nanofiber mats and natural scaffolds for sustaining growth of a liver cell line (Hep G2). Although a simple job and the novelty is just so so, the contents should be interesting to the readers of POLYMERS. I recommend its acceptance after the following issues are well addressed.

  • It should be better to give some quantitative results in your ABSTRACT.
  •  In the INTRODUCTION, it is better to give a full background about electrospinning because this journal is about “polymers” and electrospinning is an advanced polymer treatment method! For example, electrospinning is fast developing from a single-fluid blending process (. Polymers 2020, 12, 1759), to solid needle process (10.3390/polym12102421//10.1016/j.polymertesting.2020.106872), to coaxial (Polymers 2019, 11, 2008), side-by-side (10.1016/j.matdes.2020.109075), tri-axial (https://doi.org/10.3390/polym12092034) and other complex processes (10.1016/j.matdes.2020.108782). However, single-fluid process holds the biggest promise for large-scale production due to its easiness to implement.
  • “A blunt needle of 21G”, it is better to tell the readers the inner and out diameters of the metal capillaries.
  • “The scaffold thickness was measured with a digital micrometer”, how to achieve an accurate result from the soft electrospun nanofiber mats?  
  • How many times for repeating the water contact angle experiments? 116±3 is all right. Traditionally, 6 times at least.
  • Please pay attention to the references: the formats should be unified, particularly the up case and lower case of the articles’titles; the references of recent three years are too small, to relate your job with the recent developments can do favor to a high impact of your job after publication.

Author Response

It should be better to give some quantitative results in your ABSTRACT.

As suggested, certain quantitative results are given in the abstract

In the INTRODUCTION, it is better to give a full background about electrospinning because this journal is about “polymers” and electrospinning is an advanced polymer treatment method!

Thanks to the kind reviewer’s suggestion, in the Introduction section, description of electrospinning techniques has been extended, adding several very recent references (some of them from this particular journal).

“A blunt needle of 21G”, it is better to tell the readers the innerand out diameters of the metal capillaries.

In 2.1. we included the suggested needle diameters.

How many times for repeating the water contact angle experiments? 116±3 is all right. Traditionally, 6 times at least.

We included the section 2.5. Statistical analysis where we confirmed that all the measurements were done in sextuplicates.

Please pay attention to the references: the formats should beunified…

References are unified, moreover we added six new ones of more recent date.

Reviewer 2 Report

Overall comments:

The subject matter of the paper dealt with the bioactivity of electrospun PCL mats as scaffolds for HepG2 cells compared to that of liver extracellular matrix by evaluating their tensile strength, surface wettability, cell growth rate, and gene transfection efficiency.

The manuscript itself is considered to be theoretically and structurally reasonable. The present study is well worth investigating, and the authors did nicely with all the sections except for the point that there is missing meaningful ‘Discussion’ section. Above all, the authors should more elaborate upon what is of paramount importance and significance. I worry that this masks what appears to be very important subject.

Together with this critical point, there are some specific concerns that should be addressed in a point-by-point manner. If such major and minor issues were all cleared by the authors, this paper can be re-evaluated to secure its publication.

Specific concerns:

1) Arrange the contents of table 1 to be centered.

2) Regarding Fig. 1, increase the size of scale bars on the SEM images

3) Regarding Fig. 2, rearrange the spacing of two graphs which are very hard to recognize. Also, it needs to increase the font size of x-axis. Label each photo in C.

4) Unify the format of bar graphs in Figs. 2, 3, and 4. Indicate the statistically significant difference, if any, on each graph.

5) In the ‘Methods’ section, add the statistical analysis.

Author Response

Overall comments:

We agree that our results are not of greatest relevance and the topic could be tackled with more scrutiny (if affordable). However the work brings novelty in sense of systematic comparison of two cell scaffolding (bio)materials, one entirely natural (ECM) and the other synthetic (PCL). The major point of the work (and this we try to indicate in Abstract, Introdution and Conclusion) is to show that despite obvious structural/physical differences of the two materials, a relatively simple and cheap strategy (electrospun PCL) could relevantly substitute for ECM scaffolds, at certain level of bioresearch.      

In regard of the aforementioned, and in order to avoid being overly repetative, the paper was prepared with results and discussion combined in a single section, but then with obligatory Conclusion part.

Specific concerns:

  1. Arrange the contents of table 1 to be centered.

           Not clear. The data seem to be placed in the center of the table cells.

  1. Regarding Fig. 1, increase the size of scale bars on the SEM images

            Thank you for the remark. Done.

  1. Regarding Fig. 2, rearrange the spacing of two graphs which are very hard to recognize. Also, it needs to increase the font size of x-axis. Label each photo in C.

            Thank you for the remark. Done.

  1. Unify the format of bar graphs in Figs. 2, 3, and 4. Indicate the statistically significant difference, if any, on each graph.

            Thank you for the remark. Done.

  1. In the ‘Methods’ section, add the statistical analysis.

            Added as Section 2.5.

Round 2

Reviewer 2 Report

The authors sincerely provided answers to the critical issues raised on the previous review stage.
It is considered that the new version of the manuscript was well revised according to the reviewers' comments except for a couple of minor points as shown below.

Therefore, it would be accepted after minor revision.

Additionally, if it needs anything, a final English editing must be made with careful perusal just before publication.

1) Regarding Fig. 2, it is still hard to recognize it. plz. revise it.

2) Regarding Fig. 4, asterisk (*) means significant difference. So, correct "p value > 0.05" to "p < 0.05" in the caption.

Author Response

Overall comments:

Additionally, if it needs anything, a final English editing must be made with careful perusal just before publication.

Authors used InstaText, an online tool for writing and editing, to improve the text of the manuscript.

Minor comments:

  •  Regarding Fig. 2, it is still hard to recognize it. plz. revise it.

            We enlarged the font size.

  •   Regarding Fig. 4, asterisk (*) means significant difference. So,correct "p value > 0.05" to "p < 0.05" in the caption.

            Thanks for the remark. Corrected.